# Single-Input Multiple-Output (SIMO) Cascode Low-Noise Amplifier with Switchable Degeneration Inductor for Carrier Aggregation

**DOI:** 10.3390/s24206606

**Published:** 2024-10-14

**Authors:** Min-Su Kim

**Affiliations:** Department of Information and Electronic Engineering, Mokpo National University, Muan 58554, Republic of Korea; msmy970@mnu.ac.kr

**Keywords:** single-input multiple-output (SIMO), low-noise amplifier, inductive degeneration, carrier aggregation, inter-band, intra-band

## Abstract

This paper presents a single-input multiple-output (SIMO) cascode low-noise amplifier with inductive degeneration for inter- and intra-band carrier aggregation. The proposed low-noise amplifier has two output ports for flexible operation in carrier aggregation combinations for band 30 and band 7. However, during inter- and intra-band operation, gain variation occurs depending on the output mode. To compensate for this, a switching circuit is proposed to adjust the degeneration inductor, optimizing gain performance for both modes. The switching operation can minimize the control for the dynamic range in the receiver system to support carrier aggregation. The designed low-noise amplifier was fabricated using a 65 nm CMOS process, occupying an area of 2.1 mm^2^. In inter-band operation, the small-signal gain was measured by 18.9 dB for band 30 and 18.6 dB for band 7, with the noise figures of 1.03 dB and 1.07 dB, respectively. For intra-band operation, the small-signal gain was 17.3 dB and 17.2 dB, with the noise figures of 1.3 dB and 1.41 dB. The IIP_3_ values were measured by −7.6 dBm and −6.7 dBm for inter-band, and −6.3 dBm and −6.2 dBm for intra-band. Power consumption was 8.04 mW and 7.68 mW in inter-band, and 17.04 mW and 17.64 mW in intra-band depending on the output configuration.

## 1. Introduction

Carrier aggregation (CA), a key feature of the 3rd Generation Partnership Project Long-Term Evolution (3GPP LTE), enhances data rates and spectrum utilization by combining the bandwidths of 1.4, 3, 5, 10, 15, or 20 MHz to provide wideband mobile services of up to 100 MHz [1,2,3,4]. This technology has evolved with LTE-Advanced (LTE-A) Rel-14 to support scenarios involving five or more carrier components (CCs), and research on Radio Frequency Integrated Circuits (RFICs) supporting sub-6 GHz New Radio (NR) bands is actively ongoing [1]. Carrier aggregation can be categorized into three scenarios: inter-band, intra-band contiguous, and intra-band non-contiguous CA, based on the allocation of multiple frequency bands. These scenarios involve at least two component carriers (CCs), including a primary serving cell (PCell) and one or more secondary serving cells (SCell), to aggregate the bands of the same or different bandwidths and improve network throughput [2]. These CA combinations began with three in 3GPP Release 10 and have now expanded to 546 in Release 14 [3]. Consequently, RFIC receivers must adapt to support a wide range of CA combinations, leading to diverse structure solutions for flexible operation [4,5].

The first structure for carrier aggregation, which separates carriers from the digital baseband [6], offers a simple configuration with a single RF, single IF, and multiple digital chains. However, it has the drawback of limited capability to individually control the gain of each carrier, making it more suitable for contiguous CA combinations where gains are relatively uniform. Another approach involves operating the intermediate frequency (IF) chain by assigning each carrier to an IF reception channel [7]. This method requires the careful management of harmonic components from the IF local oscillator (LO) and image components due to gain and phase imbalances, often necessitating numerous filters, which increases RFIC complexity. Lastly, the multi-RF chain structure [8,9] is well suited for intra-band CA, with separate reception chains for low-band (LB), mid-band (MB), and high-band (HB) frequencies. However, this structure introduces increased complexity in the interface for multiple outputs. Consequently, designing receivers for efficient CA combinations while optimizing RF performance remains challenging due to the complexity of these circuits. RF receivers in mobile communication systems require excellent noise performance and high gain [10]. To achieve this, high-end platforms are increasingly adopting RF LNA-PA modules with integrated duplexers (LNA-PAMiD) to reduce the complexity of RFIC receivers and enhance their performance. Certainly, the addition of the LNA to the existing PAMiD is expected to increase component costs and current consumption. However, as the number of CA combinations increases, simplifying the receiver has become a more significant design issue from the RFIC perspective.

Figure 1 illustrates the front-end configuration for cellular RF communication. Figure 1a shows an example of a configuration for 2-TX/2-RX carrier aggregation. The PAMiD supports mid-band (MB) frequencies for the bands 1, 3, 4, 25, 34, 39, and 66, and high-band (HB) frequencies for the bands 7, 30, 40, and 41. It includes an antenna switch module (ASM) and duplexers to handle these bands [11]. The receiver of PAMiD generates multiple output ports to support various CA combinations with RFICs, connecting to the RFIC’s multi-input ports via a matching circuit (M/N) for impedance matching and connection lines. This interface uses a multi-layer PCB trace line, but the loss of the matching circuit and lines increases the noise performance and system complexity, impacting the receiver’s high sensitivity. For instance, if the PAMiD, including the ASM, has a receiver loss of 3.0 dB, and the matching and line loss between the PAMiD and RFIC is 1 dB, while the gain of the RFIC’s receiver LNA is 18 dB with a noise figure (NF) of 2 dB, the cascaded noise performance of the overall system will be 6 dB. However, if the RFIC’s receiver LNA achieves an NF of 1 dB, the overall noise performance improves to 5 dB. This demonstrates how the RFIC’s complex configuration influences receiver performance, adding a design challenge. Naturally, this excludes other performance degradation factors such as gain, linearity, internal chip coupling, and signal leakage within the RFIC [12,13].

Figure 1b illustrates the configuration of the LNA-PAMiD and RFIC for high-end platforms requiring superior performance. As shown in the figure, the receiver with LNA-PAMiD can select the output port for RFIC input using a flexible LNA output selection after the ASM and duplexer. This configuration facilitates a simplified interface for the RFIC, reducing potential losses that may arise from matching circuits and connection lines. Additionally, the RFIC input can be configured to match a designated receiver chain based on the separated output signal by the LNA-PAMiD, which simplifies the receiver design. In terms of receiver sensitivity, the first-stage LNA is located in the RF front-end module, not in the RFIC, and operates as a critical block in determining the receiver system’s performance. If the LNA in the LNA-PAMiD has a gain of 18 dB and a noise figure of 2 dB, the overall noise figure of the receiver system will be 5.04 dB, which is approximately 1 dB better than if the LNA were used as the first stage within the RFIC. This performance suggests that the LNA within the RFIC can be designed with a higher noise figure of 13.3 dB while still achieving a system target noise figure of 6 dB, thereby alleviating the design burden. Consequently, a design report on a multi-stage wideband LNA for RFIC has recently been published [14,15,16,17].

This paper presents the design of a cascode LNA for a single-input multiple-output application. The designed LNA features the flexibility to select multiple outputs for supporting inter-/intra-band CA scenarios. Additionally, the degeneration inductor can be switched to adjust the gain imbalance according to inter-/intra-band operation, allowing for consistent gain and low noise performance across single-output or dual-output conditions. Section 2 and Section 3 describe the operation of the LNA for CA and detail the operating principles and design considerations of inductor switching to improve gain imbalances. Section 4 presents the simulation and measurement results of the proposed LNA, while Section 5 concludes the paper.

## 2. SIMO Cascode LNA for Inter-/Intra-Band CA

Figure 2 presents the schematic of the proposed flexible output port along with the output selection operation diagram for inter- and intra-band CA scenarios. The proposed low-noise amplifier features two inputs for each high-band (HB) in a 2CC combination and an output circuit that selects the output path using common-gate transistors (M_5_, M_6_, M_7_, and M_8_). For inter-band operation, high-band signals from band 7 and band 30 are input into the core transistor M_1_ (RFIN_1_) and M_3_ (RFIN_2_), and are output through out_1_ and out_2_ via the common-gate transistor M_5_ and M_7_, respectively. Here, the common-gate transistor M_6_ and M_8_ enable flexible cross-output operation between out_1_ and out_2_. The proposed LNA incorporates a series/parallel capacitor bank to support output matching. In intra-band operation, where multiple component carriers (CCs) exist within the same band, the signals must be simultaneously split through output out_1_ and out_2_, whether the CCs are contiguous or non-contiguous. The conventional split-cascode LNA structure is recognized for its ability to easily separate output signals by utilizing a shared common-source transistor (core transistor) and two distinct common-gate transistors (cascode transistor) [16]. However, the increased current from the dual outputs leads to a non-linear gain response, as shown in Equation (1), due to the square-root dependence of transconductance (g_m_) according to the shared core transistor where $\mu_{n}$ represents the mobility of charge carriers, and $C_{ox}$ denotes the total capacitance of the transistor.
(1)$$g_{m}=\sqrt{2\mu_{n}C_{ox}\left(\frac{W}{L}\right)I_{D}}$$

As a result, the gain becomes unbalanced across the output ports. While separating the core transistor and using individual degeneration inductors could resolve these issues, it would significantly increase the chip area due to two degeneration inductors. To address this, the proposed cascode LNA has an independent core transistor for each output to ensure separate g_m_ while utilizing a shared degeneration inductor to minimize the over area. Additionally, a switchable degeneration inductor circuit is integrated to mitigate gain imbalance caused by increased current, particularly in intra-band CA scenarios.
(2)$$Z_{in,interband\;CA}\approx s\left(L_{deg}+\frac{1}{C_{gs}}\right)+\frac{g_{m}\times L_{deg}}{C_{gs}}$$
(3)$$Z_{in,intraband\;CA}\approx s\left(L_{deg}+\frac{2}{C_{gs}^{\prime }}\right)+\frac{2g_{m}^{\prime }\times L_{deg}}{C_{gs}^{\prime }}$$

Equations (2) and (3) represent the LNA input impedance for inter- and intra-band scenarios, respectively. For simplicity, the circuit impedances for input matching and the gate-to-drain capacitance (C_gd_) are omitted, and the gate-to-source capacitance (C_gs_) from the non-operating transistor is neglected. The input impedance of the LNA for inter-band CA matches that of a general cascode LNA with a degeneration inductor. However, for intra-band CA operation, both M_1_ and M_2_ are used, effectively doubling the C_gs_. Consequently, the impedance of Z_in_ for intra-band CA experiences a change twice as large as that for the input impedance of inter-band LNA, leading to performance variations. Here, C_gs_’ and g_m_’ represent the combined C_gs_ and g_m_ contributions from M_1_ and M_2_, respectively. The variation in LNA performance according to the CA combination is due to differing input impedances. To address this, the proposed LNA with a switchable degeneration inductor can adjust its input impedance by switching the impedance of L_deg_. This allows simultaneous optimization for inter- and intra-band operations, minimizing gain mismatch. However, due to the impedance differences arising from the mismatches in g_m_ and the parasitic components of layout, achieving identical impedance with an L_deg_ of 0.5 times, as indicated in the formula, is not feasible. Therefore, the optimization of L_deg_ switching is necessary.

## 3. Switchable Degeneration Inductor

Figure 3 illustrates the circuit diagram and equivalent model of the proposed switchable degeneration inductor. As shown in Figure 3a, the degeneration inductor has a tap point to optimize inductance according to the CA mode, and a switching transistor is placed between the tap point and the ground bump. This switch transistor with a degeneration inductor can be modeled with parasitic components and its equivalent model changes depending on the on/off state of the switch. Figure 3b shows the current path based on the switch’s state. During intra-band CA operation, the switch is turned on, and the inductor operates with the R_on_ resistance generated in this state, along with the composite capacitor C_TOTAL,on_, which includes C_gs,on_ and C_gd,on_. Conversely, when the switch is off for inter-band CA, it is ideally represented by infinite R_off_ and the parasitic capacitor C_TOTAL,off_. At this point, the impedance can be expressed by the following equation:
(4)$$Z_{in,Intraband}\approx jwL_{1}+\frac{R_{on}}{1+jwC_{total,on}R_{on}}$$
(5)$$Z_{in,Interband}\approx jwL_{1}+\frac{jwL_{2}}{1-w^{2}C_{totla,off}L_{2}}$$

In the case of intra-band CA, as shown in the above formula, R_on_ is minimized, allowing the inductance of L_2_ to be negligible, and the impedance can be determined primarily by L_1_. For inter-band CA, assuming R_off_ is infinitely large, if the condition of the w^2^C_TOTAL,off_L_2_ << 1 is met, the impedance will be determined by both L_1_ and L_2_.

Figure 4 presents the changes in R_on_, C_TOTAL_, and inductance according to the size of the switch transistor. Figure 4a shows the R_on_ and C_TOTAL_ values for inter- and intra-band operations. In intra-band operation, where the switching operation of the degeneration inductor is necessary, the switch transistor requires a low R_on_, which in turn demands a large-width transistor. However, a larger-width transistor increases the parasitic capacitance C_TOTAL_, which complicates achieving the optimal impedance for the desired degeneration inductance. Furthermore, the large parasitic capacitance in inter-band operation can form a series resonance ($L_{1}+\left(C_{TOTAL}\left\|L_{2}\right.\right))$ with the degeneration inductor, resulting in an undesired inductance value. Figure 4b shows that when the degeneration inductor for intra-band operation is assumed to be 0.25 nH, a switch transistor width of 200~300 um can be used. Here, this optimal inductance value is determined using simulation, and the tap point of the inductor is chosen accordingly. The proposed switchable degeneration inductor circuit can be optimized to meet the same input impedance for inter- and intra-band operations, thereby enhancing the performance of the split-cascode LNA with a shared degeneration inductor. In this design, a degeneration inductor of 0.4 nH is utilized for inter-band operation, while an inductor of 0.23 nH is employed for intra-band operation. The transistor has a width of 300 um.

## 4. Implementation and Measurement Results

Figure 5 shows the chip layout and the test-bench setup for performance verification. The LNA is designed with core transistors for band 30 and band 7, an interface, and includes the cascode transistors and load components. The proposed LNA is fabricated using a 65 nm CMOS process, and the output loads integrate the capacitor bank (series and parallel type) for output matching. The cap bank can be adjusted to achieve optimal output matching through 7-bit control in both series and parallel configurations. The total chip size is 1499 × 1776 um^2^, and the actual size of the proposed LNA, excluding 0.56 mm^2^, is approximately 2.1 mm^2^. The test bench utilized Rohde and Schwarz’s ZVA50 network and Keysight’s N8975A analyzer equipment, with the performance of the measured chip evaluated on an evaluation board designed with bump connections for flip-chip. The evaluation board designed for small-signal measurement was implemented using Roger’s RF35 substrate. The input and output ports were connected to a vector network analyzer (VNA), and noise figure evaluations were conducted using a noise source. Keysight’s E3636A was used to supply power to the designed chip.

Figure 6 presents the simulation and measurement results for the inter- and intra-band CA for band 30 and band 7. As shown in Figure 6a, the simulation results for band 30 in inter-band operation are 19.4 dB and 19.7 dB, and the measured results are 18.5 dB and 18.9 dB at output 1 and output 2, respectively. For band 7, the simulation results are 19.4 dB and 19.7 dB, and the measured results are 18.5 dB and 18.7 dB at output 1 and output 2. The corresponding noise figures are shown in Figure 6b, with band 30 having 0.89 dB and 1.03 dB for output 1, and 0.89 dB and 1.01 dB for output 2. Band 7 has noise figures of 0.88 dB and 1.07 dB for output 1, and 0.89 dB and 1.06 dB for output 2. Figure 6c shows the simulation and measurement results for intra-band operation. For band 30, the simulation gains are 17.9 dB and 18.1 dB, and the measured gains are 17.3 dB for both outputs. For band 7, the simulation gains are 17.8 dB and 18.0 dB, and the measured gains are 17.2 dB for both outports. The corresponding noise figures are shown in Figure 6d, with band 30 having 1.06 dB and 1.29 dB for output 1, and 1.04 dB and 1.30 dB for output 2. Band 7 shows noise figures of 1.04 dB for output 1, and 1.10 dB and 1.39 dB for output 2.

All the simulation and measurement results involve different control bits of the capacitor bank for each output load condition. Without the switchable degeneration inductor, the simulation results show a gain reduction from 19.4 dB to 16.5 dB due to changes in input impedance. However, the proposed LNA with the switchable degeneration inductor maintains consistent input matching conditions in both the inter- and intra-band operations, minimizing performance variation. During inter-band operation, the current consumption was 6.2 mA for each output. The current consumption was measured to be more than twice that of 14.2 mA and 14.7 mA in intra-band operation.

Figure 7 shows the measured 3rd input intercept point (IIP_3_) at a tone spacing of ±1 MHz. For band 30, IIP_3_ was measured at −30 dBm input power with tones at 2.354 GHz and 2.356 GHz, resulting in a maximum IIP_3_ of −5.9 dBm. For band 7, with tones at 2.654 GHz and 2.656 GHz, IIP_3_ was measured at −34 dBm input power, resulting in a maximum IIP_3_ of −7.6 dBm. The performance metrics of the proposed and other state-of-the-art low-noise amplifiers are listed in Table 1. The proposed LNA exhibits a lower NF performance compared to the previously studied LNAs within RFICs.

Positioned in front of the receiver, it is expected to significantly enhance the overall receiver performance with a minimum gain of 17.2 dB and excellent linearity characteristics of −6.2 dBm. In addition, according to reference [17], it is judged that the RFIC receiver for 2CC and 5CC will not increase more than twice from 2.5 mm^2^ to 3.3 mm^2^ depending on whether an LNA is added in front of the RFIC, which can help improve the complexity of the system in terms of RFIC.

## 5. Conclusions

This paper presents a low-noise amplifier (LNA) designed for a receiver with carrier aggregation that supports various combinations. The RF receiver, featuring a SIMO structure, increasingly demands a versatile RF front-end module. The proposed design includes a split-cascode LNA and a switchable degeneration inductor to minimize performance variations in both inter- and intra-band CA scenarios. The design aims to enhance reception performance and flexibility in receiver systems that require diverse CA combinations.

## Figures and Tables

**Figure 1 sensors-24-06606-f001:**
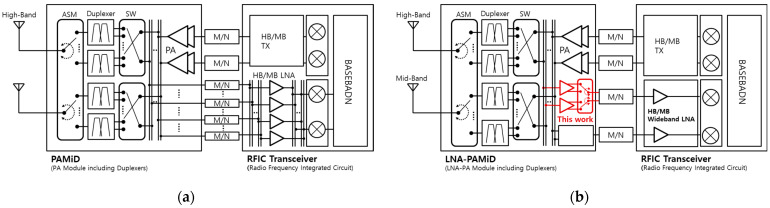
Receiver block diagram including front-end module and RFICs for high/mid-band: (**a**) with PAMiD and (**b**) with LNA-PAMiD.

**Figure 2 sensors-24-06606-f002:**
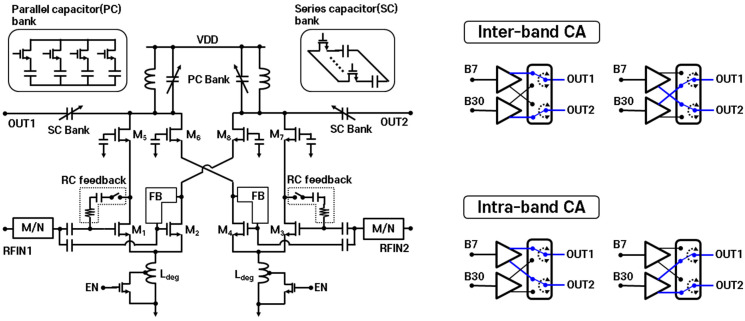
Schematic of the proposed LNA and output selection for inter-/intra-band CA.

**Figure 3 sensors-24-06606-f003:**
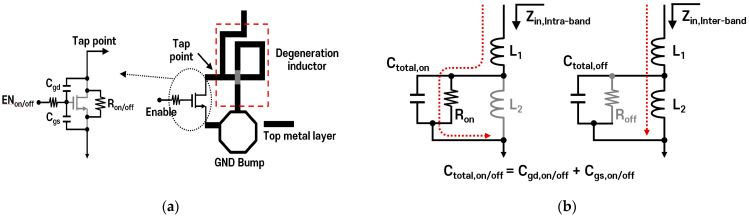
Switchable degeneration inductor: (**a**) with transistor equivalent model and (**b**) schematic according to switch on/off conditions.

**Figure 4 sensors-24-06606-f004:**
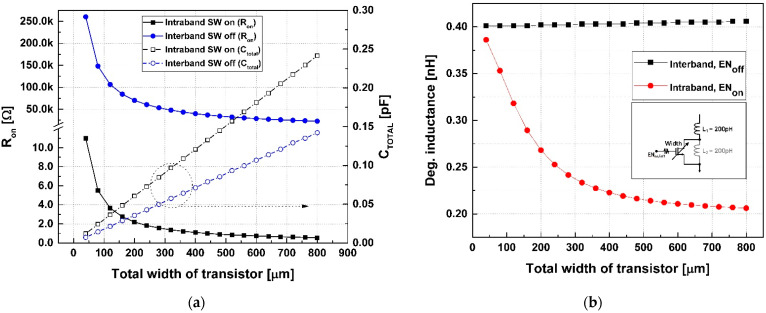
Switch on/off condition: (**a**) R_on_ and C_TOTAL_ variation and (**b**) inductance according to transistor width.

**Figure 5 sensors-24-06606-f005:**
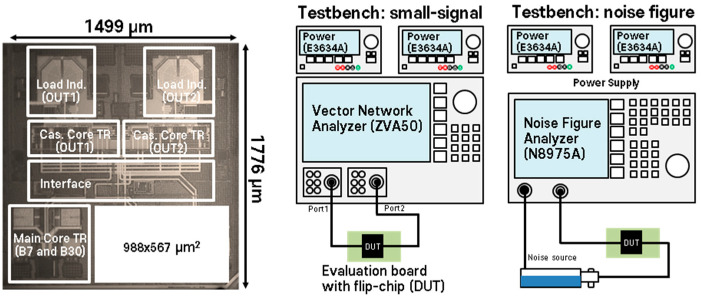
Microchip photograph and measurement setup for the proposed low-noise amplifier.

**Figure 6 sensors-24-06606-f006:**
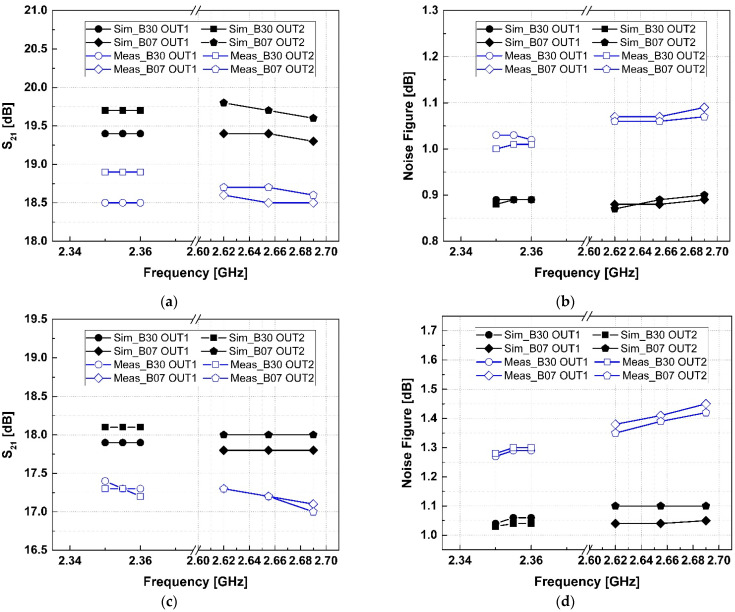
The simulated and measured results according to output ports: (**a**) S_21_, (**b**) noise figure for inter-band CA, (**c**) S_21_, and (**d**) noise figure for intra-band CA.

**Figure 7 sensors-24-06606-f007:**
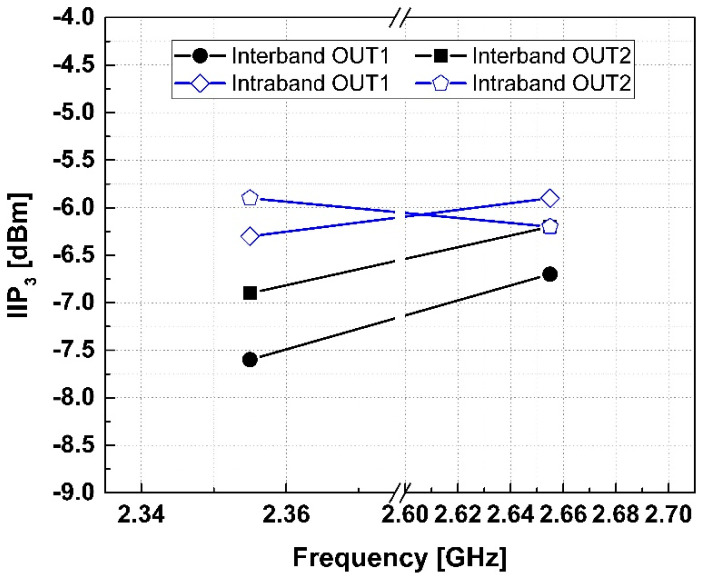
Measured 3rd input intercept point (IIP_3_).

**Table 1 sensors-24-06606-t001:** Performance comparison to previous research and products.

Ref.	Techno.	Design /Max. CC	CA Mode	Freq. [GHz]	Gain [dB]	NF [dB]	Pdc [mW]	IIP_3_ [dBm]	Size [mm^2^]
[15] JSSC’2015	45 nm CMOS	Resistor Feedback/>2	Intra-band	Around 2	37 *	<3.9 *	7 ^#^	−13.7	0.15
[16] JLPEA’2017 ^##^	65 nm CMOS	Cascode shut-off/2	Inter-band	2.62~2.69	58.9 **	1.52	11.1	−7.7	-
			Intra-band		57.1 **	1.76	21.3	−6.7	-
[17] TMTT’2021	14 nm CMOS	Wideband/5	Inter-band	0.6~2.7	62 *	4.5 *	4	−2	3.3
			Intra-band			4.9 *	10	−2	
[18] MDPI’2023	0.15 um GaN	Wideband/1	-	24.25~29.5	18.3~20.2	2.5~3.1	900	-	3.75
[19] MDPI’2024	0.25 um GaAs	Wideband/1	-	4.5~8	18	2.0	89.9	7.5	2.28 ^%^
This work	65 nm CMOS	Split-Cascode/2	Inter-band	2.35~2.36	18.9	1.03	8.04	−7.6	2.1
				2.62~2.69	18.6	1.07	7.68	−6.7	
			Intra-band	2.35~2.36	17.3	1.30	17.04	−6.3	
				2.62~2.69	17.2	1.41	17.64	−6.2	

* LNA + Mix + TIA (DSB NF); ^#^ LNA only; ^##^ simulation results; ** voltage gain; ^%^ LNA + PA + Switch.

## Data Availability

Data are contained within the article.

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
