# Peer review of "Single-Input Multiple-Output (SIMO) Cascode Low-Noise Amplifier with Switchable Degeneration Inductor for Carrier Aggregation"

_sensors, 2024, doi:10.3390/s24206606_

Round 1
Reviewer 1 Report
Comments and Suggestions for Authors
The paper presents an innovative single-input multiple-output (SIMO) cascode low-noise amplifier (LNA) that incorporates switchable degeneration inductors to optimize gain performance for inter- and intra-band carrier aggregation. The paper is well-structured and logically presented, with a clear flow from the introduction to the conclusion.
The key innovation of this work is the switchable degeneration inductor, which allows the LNA to dynamically adjust its impedance and gain for inter- and intra-band CA modes to addresses the gain imbalance.
while the innovation offers significant enhancements, it may still face challenges related to manufacturing variability and the need for precise control circuitry, which could limit its practicality in certain applications. Moreover, while the noise figure reported (1.03 dB for in-band) is competitive, the design still faces challenges related to linearity and gain imbalances that could arise from the shared degeneration inductor, which previous designs might have addressed through dedicated inductors for each output.
suggestions:
1.From the perspect of writing, the article can further elaborate on some key concepts, making it convenient for readers to read without sacrificing depth.
2.suggest to provide a detailed comparison of performance indicators between the proposed SIMO LNA and standard fixed inductor designs, highlighting specific improvements in gain and noise figure.
3. prefer to discuss trade-offs in design, such as balancing increased flexibility with potential drawbacks (like higher power consumption), and provide more context for innovation in this design.
4. It would be nice to explore how the complexity of switch circuits affects the integration of SIMO LNA with existing RF systems.
5. It is recommended to conduct a more thorough analysis on how to achieve a noise figure of 1.03 dB, starting with basic principles and formula derivation.
Comments on the Quality of English Language
The paper presents an innovative single-input multiple-output (SIMO) cascode low-noise amplifier (LNA) that incorporates switchable degeneration inductors to optimize gain performance for inter- and intra-band carrier aggregation. The paper is well-structured and logically presented, with a clear flow from the introduction to the conclusion.
The key innovation of this work is the switchable degeneration inductor, which allows the LNA to dynamically adjust its impedance and gain for inter- and intra-band CA modes to addresses the gain imbalance.
while the innovation offers significant enhancements, it may still face challenges related to manufacturing variability and the need for precise control circuitry, which could limit its practicality in certain applications. Moreover, while the noise figure reported (1.03 dB for in-band) is competitive, the design still faces challenges related to linearity and gain imbalances that could arise from the shared degeneration inductor, which previous designs might have addressed through dedicated inductors for each output.
suggestions:
1.From the perspect of writing, the article can further elaborate on some key concepts, making it convenient for readers to read without sacrificing depth.
2.suggest to provide a detailed comparison of performance indicators between the proposed SIMO LNA and standard fixed inductor designs, highlighting specific improvements in gain and noise figure.
3. prefer to discuss trade-offs in design, such as balancing increased flexibility with potential drawbacks (like higher power consumption), and provide more context for innovation in this design.
4. It would be nice to explore how the complexity of switch circuits affects the integration of SIMO LNA with existing RF systems.
5. It is recommended to conduct a more thorough analysis on how to achieve a noise figure of 1.03 dB, starting with basic principles and formula derivation.
Reviewer 2 Report
Comments and Suggestions for Authors
In this manuscript, the authors present a SIMO Cascode LNA using CMOS 65nm technology. The overall presentation is fine. I have some concerns that should be addressed before allowing for publication.
1) The grammar and English must be improved. I found many typos.
2) The number of references are not enough (only 17!)
3) In designing RF circuits, the main concern in minimized technologies is to compensate the low gm and Ion values. When it comes to the sub 30nm footprint, CNTFET technology presents better results. Why the authors did not use a better one than CMOS?
4) The authors should provide a literature review regarding the usage of emerging non-si technologies in LNAs. I recommend to use the following reference for writing this section:
https://doi.org/10.1016/j.physe.2021.114915
Comments on the Quality of English Language
There are some typos should be modified by auhors
Reviewer 3 Report
Comments and Suggestions for Authors
1. Introduction
-
Please define 3GPP LTE and expand the acronyms.
-
Please expand acronyms RFICs and NR
-
The first paragraph is not reader-friendly. “While 3GPP 36 Rel-10 defined three CA combinations, Rel-14 has expanded this to combinations”. What are Rel-10 and Rel-14? Please define these terms and their relevance to the article. Please rewrite the first paragraph in a way that readers can follow it without difficulty.
2. SIMO Cascode LNA for Inter-/Intra-band CA
-
However, the increased current from the dual outputs leads to a non-linear gain response, as the square-root dependence of transconductance (gm) according to the shared core transistor. - Please illustrate this with an equation.
-
Line 132 - Please be consistent with the subscript usage.
3. Switchable Degeneration Inductor
-
Line 165 - the condition to be met is w2CTO-165 TAL,offL2 << 1 and not 1-w2CTO-165 TAL,offL2 << 1. Please correct this.
-
Figure 4(b) - Please explain why the interband inductance increases with the total width of the transistor whereas the intraband inductance decreases drastically with the total width?
-
Line 173 - Please define the required Ron and Ctotal quantitatively. What is the maximum Ron and maximum Ctotal that the circuit can tolerate?
4. Implementation and measurement results
-
Please elaborate on the measurement setup demonstrated in Figure 5.
-
Line 215 - S11 is not the correct representation for input impedance. Please correct this.
-
The performance metrics shown in Table 1 need to be explained properly. How does your design compare with the state-of-the-art designs when it comes to gain, NF, and IIP3? It looks like other designs have achieved lower sizes compared to the design presented in this study. Please explain why. What are the merits and demerits of your design?
Comments on the Quality of English Language
Please define all the acronyms at their first appearance. Please be consistent with the subscripted symbols.
Round 2
Reviewer 3 Report
Comments and Suggestions for Authors
Good work. Thanks for addressing the comments.
Author Response
Your review comments were very helpful in my paper writing. Thank you again.